# Effect of Alkaline Conditions on Forming an Effective G4.0 PAMAM Complex with Doxorubicin

**DOI:** 10.3390/pharmaceutics15030875

**Published:** 2023-03-08

**Authors:** Magdalena Szota, Barbara Jachimska

**Affiliations:** Jerzy Haber Institute of Catalysis and Surface Chemistry, Polish Academy of Sciences, 30-239 Krakow, Poland

**Keywords:** PAMAM dendrimers, doxorubicin, drug delivery systems, nanotechnology, PAMAM-DOX complex

## Abstract

In this study, special attention was paid to the correlation between the degree of ionization of the components and the effective formation of the complex under alkaline conditions. Using UV-Vis, ^1^H NMR, and CD, structural changes of the drug depending on the pH were monitored. In the pH range of 9.0 to 10.0, the G4.0 PAMAM dendrimer can bind 1 to 10 DOX molecules, while the efficiency increases with the concentration of the drug relative to the carrier. The binding efficiency was described by the parameters of loading content (LC = 4.80–39.20%) and encapsulation efficiency (EE = 17.21–40.16%), whose values increased twofold or even fourfold depending on the conditions. The highest efficiency was obtained for G4.0PAMAM-DOX at a molar ratio of 1:24. Nevertheless, regardless of the conditions, the DLS study indicates system aggregation. Changes in the zeta potential confirm the immobilization of an average of two drug molecules on the dendrimer’s surface. Circular dichroism spectra analysis shows a stable dendrimer-drug complex for all the systems obtained. Since the doxorubicin molecule can simultaneously act as a therapeutic and an imaging agent, the theranostic properties of the PAMAM-DOX system have been demonstrated by the high fluorescence intensity observable on fluorescence microscopy.

## 1. Introduction

Doxorubicin (DOX) is a chemical compound that belongs to the group of anthracycline antibiotics, in which the planar anthraquinone nucleus of the molecule is connected by a glycosidic bond, at the ring atom 10, to the amino sugar called daunosamine (Figure 1b) [1]. Doxorubicin is highly effective in treating acute leukemias and lymphomas, sarcomas, bladder, breast, lung, ovarian, gastric, thyroid, and many other types of cancer [1,2]. An extremely important feature of the structure of this molecule is internal fluorescence, which is a valuable characteristic for diagnostics and imaging [3,4].

A known complication of anthracycline-based therapy is dose-dependent cardiotoxicity, which can be fatal [5,6]. Due to the fact that cardiotoxicity may also develop after discontinuation of therapy and cardiac damage may occur after the first dose, the therapeutic dose of anthracyclines has been limited to 450–500 mg/m^2^ of body weight [6]. Van Hoff et al. showed that with the culminating DOX doses of 400, 550, and 750 mg/m^2^, the incidence of cardiotoxicity increases and amounts to 3.7% and 18%, respectively. Therefore, it is recommended that the cumulative dose of doxorubicin should not exceed 550 mg/m^2^ [7,8]. One way to minimize cardiotoxicity may be by using a nanocarrier-based DOX delivery system (DDS), which can alter the pharmacological distribution, ultimately reducing its adverse accumulation in the heart [9].

The polymers used in DDS systems are dendrimer nanoparticles. Poly(amidoamine) dendrimers (PAMAM) consist of many monomeric branches connected in the central part by a core, present in the form of ethylenediamine [10,11]. Branches from the core may be terminated with amino, hydroxyl, or carboxyl groups [12]. The main advantages of dendrimers are their well-defined globular structure, monodispersity, and polyvalency. The size of PAMAM dendrimers can range from 1–100 nm and depends on the generation (G0, G1, G2, G3, etc.), defined as the number of the next inner layer consisting of branches extending from the core and built of repeatable units [11,12,13,14]. The dendrimer’s generation, and thus the size and number of functional groups, determine its physicochemical parameters and ability to encapsulate the drug [15]. A highly attractive feature of dendrimer molecules is the possibility of extensive modification of surface groups and, thus, the ease of creating conjugates with selected ligands. In addition, dendrimers, due to their amphiphilic nature, can increase the solubility and bioavailability of both hydrophobic and hydrophilic drugs [12]. Considering the hydrophobic core of PAMAM dendrimers and the hydrophilic nature of the peripheral part, a structural analogy to micelles can be observed, which have the ability to both ionically and hydrophobically bind ligands [16,17]. Active molecules can be transferred using dendrimer systems: by immobilizing the surface structure or encapsulating inside. Dendrimers can transport ligands that are bound in various ways, which can affect the system’s stability and release kinetics. These strategies include physical as well as chemical interactions. The first contains non-covalent association, hydrogen bonding, hydrophobic, or electrostatic interactions, while the second involves the covalent coupling of drugs to surface groups of dendrimers [18].

Various nanosystems, such as liposomes, dendrimers, polymers, micelles, nanogels, and metallic nanoparticles, are used for the targeted delivery of DOX [9,15,19]. The currently commercially available formulations for doxorubicin delivery approved by the FDA are Doxil^®^ and Myocet^®^, both based on liposomal nanocarriers [9]. The downside of nanotechnology-based therapy based on liposome systems is the lack of targeting, poor biodistribution, and low blood-brain barrier penetration [19]. In addition, liposomal DOX can cause skin reactions caused by the hydrophilic coating of the liposomes used to increase the permeation of the system through the skin layers and thus lead to the absorption of DOX in the stratum corneum [19]. Dendrimers are more stable due to their unique cationic end groups, but this stability may increase hemolytic toxicity, especially with higher-generation PAMAM dendrimers (>G5 PAMAM) [12,14,19]. In addition, the nanosize of the dendrimers and their interaction with plasma proteins allow them to drive passive targeting through increased permeability and retention (EPR) [12]. The above-mentioned advantages resulting from the unique structure of dendrimers may play a key role in the controlled and targeted release of doxorubicin and thus reduce its cardiotoxicity. A DOX-conjugated G5-NH_2_ dendrimer-based carrier with a diaminobutane core was shown by Kuruvilla et al. to increase anticancer activity and reduce DOX-induced cardiotoxicity in a mouse hepatocellular carcinoma (HCC) model [20]. Kaminskas et al. documented a similar effect in a preparation based on a 5th-generation PEG-ylated polylysine dendrimer, which was characterized by good efficacy and showed a much lower risk of cardiotoxicity than in the case of the drug alone [21]. Researchers mainly focus on obtaining dendrimer-DOX complexes under physiological conditions, often conjugated to other ligands and/or linkers, such as amides, hydrazone, folic acid, hyaluronic acid, or peptides [15,22,23,24]. Guo et al. presented a combination therapy using doxorubicin and cisplatin conjugated with G4.0 PAMAM and hyaluronic acid, which showed high efficacy considering the inhibition of tumor growth as well as the reduction of DOX toxicity [24]. Zhong et al. demonstrated an equally high efficacy of DOX conjugate with carboxylated G4.0 PAMAM in local therapy of lung cancer [25,26]. Many studies show that dendrimers have great potential in active substance delivery and imaging. In contrast, they have yet to be tested in clinical trials for cancer therapy [15].

Despite the many studies on various dendrimer conjugates, the present study focuses on the interaction in the G4.0 PAMAM-DOX system at the molecular level. In the present study, special attention was paid to the effect of the degree of ionization of the components on the efficiency of complex formation, so a method of forming G4.0 PAMAM-DOX complexes under alkaline conditions was proposed. The following studies focused on optimizing the conditions for forming the G4.0 PAMAM complex with doxorubicin, which can affect the subsequent application of nanosystems in vitro and the kinetics of drug release. To control the degree of ionization of the system and its stability, UV-Vis, NMR, and CD spectroscopy were used. The size and effective charge of the complex were monitored using DLS and electrophoretic mobility. Images from fluorescence microscopy confirmed the monomeric form of the drug in the structure of dendrimer complexes.

## 2. Materials and Methods

### 2.1. Materials

Poly(amidoamine) dendrimer generation 4.0 (G4.0 PAMAM) was acquired from Dendritech, Inc. (Michigan; Midland, MI, USA), doxorubicin hydrochloride (DOX) from Ambeed (Arlington, IL, USA) and deuterium oxide (D_2_O, 99.9 atom % D) was supplied by Deutero GmbH (Kastellaun, Germany). All solutions were prepared in deionized water, with the pH adjusted using sodium hydroxide (NaOH) and hydrochloric acid (HCl) solutions.

### 2.2. UV-Vis Spectroscopy

The Thermo Scientific Evolution 201 UV-Vis spectrophotometer was used to determine the ionization degree of doxorubicin, the concentration of doxorubicin bound to the dendrimer, and the binding constants of complexes. All UV-Vis spectra were measured in the wavelength range of λ = 190–800 nm.

### 2.3. Nuclear Magnetic Resonance Spectroscopy (^1^H NMR)

The concentration of the doxorubicin sample used for NMR measurements was 1 mg/mL. All solutions were prepared in deionized water and deuterium oxide in a 1:1 ratio. The pH was adjusted using, for each value, about 15–35 μL of sodium hydroxide (NaOH) with a concentration of c = 0.1 M. The final pH of the solutions was 5.6 (native solution), 7.5, 8.5, 9.0, 9.5, and 10.0. The ^1^H NMR experiments were performed at 298 K on a Bruker Advance 600 MHz spectrometer.

### 2.4. Preparation of G4.0 PAMAM-DOX Complexes

G4.0 PAMAM complexes with doxorubicin (G4.0 PAMAM-DOX) were formed as aqueous solutions for a constant dendrimer concentration of 0.25 mg/mL (17.6 µM). To study the effect of pH, the carrier–drug molar ratio was 1:6, and the initial pH values of the complexes were 9.0, 9.5, and 10.0. The carrier–drug molar ratio was varied from 1:6 and 1:12 to 1:24 at pH 9.5. The pH was adjusted using 5–40 μL of sodium hydroxide (NaOH) with a concentration of 0.1 M. The G4.0 PAMAM–DOX complexes were mixed for 24 h in darkness at room temperature (298 K).

### 2.5. Dialysis

The complexes were dialyzed using Slide-A-Lyzer™ Dialysis Cassettes (MWCO 10.0 kDa), Thermo Fisher (Waltham, MA, USA), to remove drug molecules that were not bound to the carrier. The dialysis process was carried out in distilled water at an appropriate pH. Dialysis was performed for 24 h at room temperature (298 K).

### 2.6. Dynamic Light Scattering (DLS)

The hydrodynamic diameters of the G4.0 PAMAM-DOX complexes were determined by the dynamic light scattering method using a Malvern Nano ZS analyzer.

### 2.7. Fluorescent microscopy

The one drop of the G4.0 PAMAM-DOX complexes after dialysis (molar ratios of 1:6 and 1:12 at pH 9.5) was applied to a clean mica surface and visualized using a ZEISS Axio Imager 2 Research Microscope (ZEISS International, Oberkochen, Germany).

### 2.8. Scanning electron microscopy

A drop of G4.0 PAMAM water solution with a 1 × 10−3 mg/mL concentration was applied to the carbon grid (EM Resolutions Ltd., Keele, Staffordshire, UK). After 1 min, the excess solution was drained by gently touching the edge of the grid with a filter. The grid was dried in the air for 30 s. Then the dendrimers were stained by applying a drop of uranyl acetate solution with a concentration of 1% for 2 min. After this time, the excess dye was drained, and the prepared grid with dendrimers was allowed to dry. SEM studies were carried out using a Field Emission Scanning Electron Microscope JEOL JSM-7500F equipped with an X-ray energy dispersive (EDS) system.

### 2.9. Determination of the binding constant (K_UV-Vis_) of the G4.0 PAMAM-DOX complex

The determination of pH-dependent binding constants for complex formation was performed by UV-Vis spectroscopy. A solution of DOX in molar ratios of 1:3, 1:6, 1:9, 1:12, 1:15, 1:18, 1:21, and 1:24 was added to a solution of G4.0 PAMAM with a constant concentration of 17.6 μM (0.25 mg/mL). UV-Vis spectra were measured in the wavelength range of 190–800 nm. Binding constant values were calculated using the Hill method [27].

### 2.10. Laser Doppler Velocimetry (LDV)

The electrophoretic mobility (*µ_e_*) was determined using a Malvern Nano ZS analyzer. Measurements were made in the pH range of 2.0–11.0 for aqueous solutions of pure G4.0 PAMAM and G4.0 PAMAM–DOX complexes with a dendrimer concentration of 1 mg/mL (70 µM). Electrophoretic mobility of the complexes was measured as a function of the pH of complex formation (9.0, 9.5, and 10.0) and G4.0 PAMAM-DOX molar ratios (1:3, 1:6, and 1:12). The pH was adjusted using sodium hydroxide (NaOH) with a concentration of 0.1 M and hydrochloric acid (HCl) with a concentration of 0.05 M.

### 2.11. Circular Dichroism (CD)

Circular dichroism was used to analyze changes in the structure of doxorubicin hydrochloride in water for pH 5.6–11.0. The concentration of the DOX was 0.1 mg/mL. CD spectra for G4.0 PAMAM-DOX were recorded for complexes at a molar ratio of 1:6 and pH 9.5, before and after dialysis. A Jasco-1500 spectrometer (Jasco, MD, USA) with a 10 nm quartz cuvette was used for the measurements. The spectra were recorded in the wavelength range 185–300 nm with a resolution of 1 nm, while the scanning speed was equal to 50 nm/min.

## 3. Results

### 3.1. Degree of Ionization of the Doxorubicin Molecule Depending on the pH of the Solution

The doxorubicin molecule comprises a planar anthraquinone nucleus linked to an amino sugar via a glycosidic bond [1]. The DOX molecule contains functional groups such as the phenolic ring, acidic groups, and basic functional groups in the amino sugar. This makes it both amphiphilic and amphoteric. According to literature data, doxorubicin in an aqueous solution is characterized by different ionization states defined by pKa constants of 8.2 and 9.5. Below pH 8.2, it occurs in the form of a cation, between pH 8.2 and 9.5 in a neutral form, while at pH above 9.5, the anionic form predominates [28,29]. Other sources report that the deprotonation of daunosamine (-NH_3_) and one of the phenolic (-OH) groups of the B aglycone ring in aqueous media occur at pK values of approximately 8.15 and 10.16 [30], respectively. The UV-Vis spectroscopy method was used to determine changes in drug ionization. DOX absorption spectra are characterized by the presence of two main bands of high intensity, around λ = 290 nm and λ = 480 nm, which are assigned to the π → π* transitions polarized along the short (z) and long (y) axes of the drug molecule, respectively [30,31]. The peak corresponding to polarization along the short axis is shifted from 290 nm to 300 nm due to the self-association of doxorubicin molecules in an aqueous solution [22]. The band at λ = 480 nm associated with polarization along the long axis is highly susceptible to most structural changes, including association processes and changes in the molecule’s ionization. Therefore, changes occurring in this range allow observation of changes in the drug molecules form [31]. With increasing pH, the absorbance successively decreases, the positioning of the bands undergoes a bathochromic shift, and the most significant change in the spectra is visible above pH 9.26 (Figure 2). The observed spectral shifts are attributed to the deprotonation of the hydroxyl group (-OH) at the C(11) carbon (*pK_a_* = 10.16) [32].

The course of changes in the absorbance value and the position of the maximum are shown in Figure 3a. The control of shifts in the position of the spectral maximum depends on the degree of ionization of the substance. It allows the determination of the transition conditions between individual forms. Due to the significant changes in the spectrum, calibration curves are made for individual pH conditions to determine the drug concentration in the complexes. Figure 3b shows the calibration curve for DOX in an aqueous solution of pH 7.5. The absorbance for DOX increases linearly in the concentration range of 1.8–184 μM (0.001–0.1 mg/mL) and is characterized by an extinction constant at *λ* = 481 nm equal to *ε*_7.5_ = 9610 M^−1^ cm^−1^. Ngyuen et al. showed that the extinction coefficient for an aqueous DOX solution is higher and amounts to *ε* = 13,500 ± 100 M^−1^ cm^−1^ for native pH [33]. This is consistent with the tendency for changes in absorbance depending on pH. Absorbance at native pH is higher than physiological pH (Figure 3a).

Figure 4 shows the 600-MHz ^1^H NMR spectrum for DOX as a function of the degree of ionization of the molecule in the pH range 5.6–10.0, where 5.6 is the native pH for the drug. The ^1^H NMR spectra for DOX are characterized by peaks in the range of 1–8 ppm [34]. Due to the performance of the spectra at an H_2_O/D_2_O ratio of 1:1 and the large amount of water in the system, the signal at 4.5 ppm was lost. As the pH increases above 8.5, signals from most protons and upfield shifts of peaks at 3.5 ppm are gradually disappearing. These changes may be indicative of protonation and structural changes in DOX.

### 3.2. Influence of pH on the efficiency of G4.0 PAMAM-DOX complex formation

The efficiency of complex formation was based on UV-Vis spectroscopy and measurement of the zeta potential (ζ). The measurement series concerned complexes formed under pH conditions when doxorubicin is present in the anion form (pH 9.0, 9.5, and 10.0). The UV-Vis spectra of individual complexes before and after dialysis are presented in Figure 5.

Due to the drop in pH of the complex after dialysis from initial values (pH 9.0–10.0) to 7.0–7.3, the concentration of the drug was determined, considering the change in the extinction constant ε_DOX7.5_ = 9610 M^−1^cm^−1^. The concentration of doxorubicin in the complexes is presented in Table 1. The loading content (LC) values, i.e., the ratio of the mass of the drug incorporated into the carrier to the mass of the carrier, and the encapsulation efficiency (EE), i.e., the ratio of the mass of the drug incorporated into the carrier to the initial mass of the drug, were also determined for the complexes, according to the following formulas:(1)$$\mathrm{LC}=\frac{Amount\;of\;drug\;in\;the\;nanoparticles\;[mg]}{Amount\;of\;nanoparticles\;[mg]}\times 100\%$$
(2)$$\mathrm{EE}=\frac{Amount\;of\;drug\;in\;the\;nanoparticles\;[mg]}{Amount\;of\;used\;drug\;in\;initial\;stage\;[mg]}\times 100\%$$

The efficiency of DOX binding to the G4.0 PAMAM dendrimer strongly depends on the degree of ionization of the components, as in the case of 5-fluorouracil [35,36]. The highest efficiency occurs at pH 9.0 and 9.5, where DOX concentrations are 21 and 19 ppm, respectively. In the case of pH 10.0, the efficiency is the lowest and amounts to 12 ppm. The drug binding efficiency to the carrier structure strongly depends on the degree of ionization of both components. However, at pH 10.0, the G4.0 PAMAM dendrimer has a neutral charge on the surface, which may limit DOX binding.

### 3.3. Efficiency of G4.0 PAMAM-DOX Complex Formation Depending on the Molar Ratio of the Components

In further studies, the influence of the molar ratios of the components on the efficiency of complex formation was verified. The UV-Vis spectra of individual complexes before and after dialysis are shown in Figure 6. The concentration of doxorubicin in the complexes was calculated in the same way as in the case of measurements, depending on the pH of the solution. As the molar ratio of G4.0 PAMAM-DOX increases, we observe a gradual increase in the number of drug molecules in the complex structure (Table 2). With increasing DOX concentrations (G4.0 PAMAM-DOX molar ratios of 1:6, 1:12, and 1:24), we obtained 0.019 mg/mL, 0.047 mg/mL, and 0.098 mg/mL of the drug in the complex, respectively. Based on the content of the carrier molecule, this gives about 1.9, 4.6, and 9.6 DOX molecules bound to one G4.0 PAMAM molecule in the cases of 1:6, 1:12, and 1:24 molar ratios, respectively (Table 2). These results indicate that binding efficiency increases with ligand concentration.

Binding efficiency was estimated, which shows that encapsulation efficiency increases from 32.76 to 40.16%, while loading content increases from 7.60 to 39.20% with DOX concentration. However, our observations suggest that a specific DOX concentration limit should not be exceeded, as there is a precipitate for the complex at a molar ratio of 1:24. This may be related to the tendency of DOX to aggregate strongly at high concentrations [28,37]. Previous studies assessed the effectiveness of forming the G4PAMAM complex with 5-fluorouracil (G4-5FU) [35,36]. These studies have shown that a dendrimer molecule binds around 20 5FU molecules, and the binding efficiency strongly depends on the degree of ionization of both components [36]. Comparing the degree of loading of both drugs (LC), the efficiency of doxorubicin is about two times higher than that of 5-fluorouracil, for which LC = 18%. The encapsulation efficiency (EE) is also much higher in the case of G4-DOX in a molar ratio of 1:24 and is found at a level of 40.16%. The low EE value in the case of G4-5FU is due to the need to use during preparation a large excess of the drug in relation to the carrier (1:800). In the case of doxorubicin, it is not necessary to use a large excess of the drug relative to the number of carrier molecules. Therefore, preparing the complexes is characterized by lower losses in the dialysis stage to separate unbound drug molecules. Similar effectiveness in the case of DOX was obtained by Chanphai et al., who studied the effectiveness of drug binding to the G4.0 PAMAM structure in an aqueous solution at physiological pH (pH = 7.5). Using fluorescence spectroscopy, they estimated drug loading efficiency at 40% [38]. Chittasupho et al. examined the effectiveness of doxorubicin encapsulation in the G4.0 PAMAM structure, which was additionally surface-modified with the LFC131 peptide, which recognizes the receptor present on the surface of breast cancer cells. They achieved encapsulation efficiency and load capacity of 92.37 ± 1.03% and 33.98 ± 0.01%, respectively [39]. Zhong et al., using NMR and MALDI techniques, estimated the number of about 3.5 DOX particles bound to a G3.0 PAMAM dendrimer [25]. The efficiencies we obtained for most complexes are close to the values reported in the literature. Most researchers synthesize PAMAM-DOX complexes under physiological conditions. However, we noted that both protonation of the molecule and the use of a significant excess of the drug relative to the carrier could result in a significant improvement in binding efficiency.

### 3.4. Size and Stability Analysis of G4-DOX Complexes

The size of the G4.0 PAMAM-DOX complexes depends on the condition of the complex formation, which was determined using the DLS method. The results are presented in Table 3. The particle size distribution of G4.0 PAMAM-DOX complexes by intensity is shown in Appendix A. In all conditions, we observe the aggregation of dendrimer molecules in the presence of the drug in the system. The G4.0 PAMAM dendrimer’s hydrodynamic diameter equals d = 4.9 ± 0.1 nm at pH 10.0. The sizes of the aggregates range from 136.1 ± 6.3 (G4.0-DOX 1:24, pH 9.5) to as much as 712 ± 77.2 nm (G4.0-DOX 1:6, pH 9.0). Nevertheless, to verify the application use of the obtained polydisperse G4.0 PAMAM-DOX complexes, MTT tests were carried out, which showed the absence of cytotoxicity of the dendrimer itself and a decrease in cell viability after 72 h incubation by 62∓3% and 64∓1% for lung cancer cells (NCL-H23) and human keratinocytes (HaCaT), respectively. The cytotoxicity of the drug itself under these conditions is comparable. The use of the carrier did not result in the drug’s effectiveness, but in vitro studies require further verification.

SEM images of the G4.0 PAMAM dendrimer confirm a nanoscopic particle size of the dendrimer (Figure 7a). The picture also shows larger clusters, resulting from the aggregation of molecules on the adsorption surface. The internal fluorescence of doxorubicin was used to visualize the G4.0 PAMAM-DOX complexes, which is a very valuable system imaging tool. Doxorubicin exhibits emissions at λ = 595 nm with excitation at λ = 470 nm [40,41]. The emission wavelength of doxorubicin includes both green and red light, with green light predominating [41]. Fluorescence microscopy images show the presence of smooth and spherical nanostructures of various sizes (Figure 7b). These results confirm the aggregation tendencies of the system observed in DLS. Doxorubicin molecules result from electrostatic cross-linkage between the positively charged amino group of G4.0 and the anionic group of DOX, leading to the accumulation of excess DOX on the surface of nanocarriers and an increase in particle size. Doxorubicin occurs in the complexes as monomers because, as a dimer, it is characterized by a practically complete lack of fluorescence (Φ_DIM_ = 10^−5^). DOX monomers have a very high fluorescence yield, Φ_MON_ = 3.9 × 10^−2^ [42,43]. The phenomenon of fluorescence in G4.0 PAMAM-DOX systems not only confirms the presence of the drug in the carrier structure but also proves that it exists in a monomeric form.

### 3.5. Determination of the Doxorubicin Binding Constant K_G.0-DOX_

The shifts observed in the UV-Vis spectra of the doxorubicin molecule and the G4.0 PAMAM-DOX complex indicate the presence of interactions between the doxorubicin and the dendrimer. Based on UV-Vis spectroscopy spectra for complexes formed at pH 9.0, 9.5, and 10.0 in the DOX concentration range of 53–430 μM, the titration data were fitted to this model using the nonlinear regression method within the SigmaPlot Version 10.0 software. The DOX binding constants to the G4.0 PAMAM molecule were determined using the Hill equation [27,44,45]:(3)$$\alpha =\frac{∆A}{{∆A}_{max}}=\frac{K_{a}{\left[L\right]}_{0}^{n}}{1+K_{a}{\left[L\right]}_{0}^{n}}=\frac{{\left[L\right]}_{0}^{n}}{{(1/K}_{a})+{\left[L\right]}_{0}^{n}}$$
where ∆*A* (*= A_obs_ – A*_0_) is the change in absorbance, ∆*A_max_* is the maximum change in absorbance, [*L*]_0_ is the initial guest concentration, *n* is the Hill coefficient, and *K_a_* is the association constant [44].

The *K_UV-Vis_* binding constants were calculated to be *K*_9.0_ = (2.39 ± 0.43) × 10^2^ M^−1^, *K*_9.5_ = (2.73 ± 0.16) × 10^2^ M^−1^, and *K*_10.0_ = (3.98 ± 0.48) × 10^2^ M^−1^ for pH 9.0, 9.5, and 10.0, respectively. The value of the binding constant refers to the drug concentration needed for effective binding to the nanoparticle. Therefore, the lower the binding constant value, the greater the affinity between the drug and carrier. The obtained *K_UV-Vis_* values indicate greater binding efficiency in the case of complexes formed at pH 9.0 and 9.5. At pH 10.0, the binding constant has a higher value, which indicates a lower DOX affinity for the G4.0 PAMAM dendrimer. An identical correlation was obtained considering the encapsulation efficiency, which is 20.7% for pH 10.0, 32.8% for pH 9.5, and 36.2% for pH 9.0. Chanphai et al. determined the binding constant for the G4.0 PAMAM-DOX system using fluorescence spectroscopy under physiological conditions, and it was *K =* 1.6 × 10^6^ M^−1^ [38]. Sanyakamdhorn et al. reported the same value for the G4.0 PAMAM-DOX complex and a lower value of *K* = 1.1 × 10^5^ M^−1^ for m-PEG-G3PAMAM-DOX [46]. The binding constants determined in the submitted studies are lower, which confirms the higher affinity of DOX to the PAMAM dendrimer under alkaline conditions. Doxorubicin binding to chitosan nanoparticles with different molar masses (15, 100, and 200 kDa) is described by the constants *K* = 8.4 × 10^3^ M^−1^, *K* = 2.2 × 10^5^ M^−1^, and *K* = 3.7 × 10^4^ M^−1^, respectively, and is correlated with the particle size [47]. According to Bulavin et al., one macromolecule of bovine serum albumin (BSA) binds two DOX molecules with a binding constant of *K* = 4.79 × 10^6^ M^−1^ [48]. Comparing the obtained data with the literature values, it turns out that DOX has a higher affinity for the G4.0 PAMAM dendrimer under alkaline conditions than under physiological conditions. Much higher affinity was also obtained for other types of nanocomplexes, such as chitosan nanoparticles or BSA (Figure 8).

### 3.6. Zeta Potential (ζ) of G4.0 PAMAM-DOX Complexes

Changes in the zeta potential of the G4PAMAM dendrimer in the presence of doxorubicin depending on the conditions of complex formation are shown in Figure 9a. The addition of doxorubicin neutralizes the positive charge of the dendrimer, which is derived from the primary and tertiary amines present in the structure of the molecule. The zeta potential decreases by an average of 25% in the pH range of 3.0 to 7.5. Above the isoelectric point, the presence of the drug reduces the negative charge of G4PAMAM by an average of 45%. In addition, the isoelectric point of the complex relative to the original dendrimer is shifted from pH 9.95 toward pH 9.65 (Table 4).

The effective charge of a molecule (*N_c_)* is defined as the average number of charges present in one molecule and has been calculated using the Stokes-Lorenz equation [49,50]:(4)$$N_{c}=\frac{6\pi \eta R_{H}}{e}\mu_{e}$$
where *e* = 1.602·10^−12^ C and *η* = 8.9·10^−3^ g·(cm·s)^−1^ [51].

The effective charge of the G4PAMAM dendrimer molecule in water is the highest at pH = 6.3 and amounts to 11.7e. Based on the *N_c_* value, the degree of ionization (α_e_) was determined, defined as the ratio of *N_c_* to the nominal charge of the molecule *N_m_* (*N_m_* = 64 for G4PAMAM) [49]. The maximum degree of ionization αe for G4PAMAM is 19.0% at pH = 6.3. This means that at this pH value, twelve surface groups are protonated, to which a maximum of 12 drug molecules can attach as a result of electrostatic interactions. At pH = 7.0, a dendrimer molecule on the outer surface can attach a maximum of eleven molecules (*α_e_* = 17.8%), and at pH = 10, only about two molecules (*α*_e_ = 3.0%). Considering the degree of ionization of the G4PAMAM molecule in the entire pH range and the neutralization of the surface charge as a result of complexation with doxorubicin, it turns out that a minimum of two drug particles electrostatically interact with the dendrimer surface amino groups. Comparing this data with the results obtained using UV-Vis spectroscopy shown in Figure 4 and Figure 5 for complexes formed at pH 9.0, 9.5, and 10.0 (at the molar ratio of 1:6), two doxorubicin molecules are immobilized on the surface of G4PAMAM. An exception are the complexes with PAMAM-DOX molar ratios of 1:12 and 1:24 formed at pH 9.5, where approximately five and ten DOX molecules accumulate on one dendrimer molecule, respectively. It should be noted that inside the G4PAMAM structure there are as many as 62 functional groups, which are all potentially active centers for drug binding. Considering the results of the zeta potential measurement in relation to the UV-Vis results, we can assume that some of the doxorubicin particles are encapsulated inside the carrier structure. A similar effect was observed with the dendrimer complex with 5-fluorouracil, which compensated the charge of G4PAMAM by about 22% [35].

### 3.7. Stability of G4.0 PAMAM-DOX Complexes Using Circular Dichroism (CD)

Compounds from the anthracycline group, including doxorubicin, are characterized by the presence of several asymmetric carbon centers in their structure, that enable structural analysis using circular dichroism [30]. Doxorubicin has a chromophore with numerous π → π* and n → π* transitions. Therefore, in both electron absorption and CD spectra, it is possible to control the chromophore deprotonation process or bind other biological molecules or metal ions [31,32]. Monitoring of the potential changes occurring within the structure of doxorubicin during complexation with the G4PAMAM dendrimer and the related role of pH was carried out using the CD method in the near ultraviolet range of 185–300 nm (Figure 10). The CD spectrum of the DOX molecule at native pH (pH = 5.6) and physiological pH is characterized by the presence of a peak maximum at *λ* = 202 nm and two minimum peaks at *λ* = 233 nm and *λ* = 252 nm (Figure 10a, blue and pink lines). With the increase in pH in the range of 8.5–10.0, a bathochromic shift of the peaks is visible, also observed in the UV-Vis spectra, confirming the gradual deprotonation of the molecule. Above pH 10.0, a sudden decrease in spectral intensity is observed, which may be related to the degradation of the DOX molecule in strongly alkaline conditions as a result of cleavage of the glycosidic bond and/or breaking of the A ring, which in turn leads to the asymmetry of the molecule and a decrease in the intensity of the CD spectra [31].

The CD spectra for the G4.0 PAMAM-DOX complexes, depending on the molar ratio, are shown in Figure 10b. Changes in the intensity of the peaks are adequate for the molar ratio of the complexes; the higher the concentration of DOX in the complex, the higher the intensity of the spectrum. In addition, the spectra of the complexes before and after dialysis were compared. In the case of a complex with a molar ratio of 1:6, no shift in the position of the spectral peaks after dialysis was observed. In the case of 1:12, a shift in the position of the peaks towards longer wavelengths was indicated. For the 1:24 complex, the changes are toward shorter wavelengths, probably due to the large aggregation observed in this case and changes in the pH range of the complex itself.

## 4. Conclusions

The study monitored the effects of the ionization degree of doxorubicin on the effectiveness of complex formation with the G4.0 PAMAM dendrimer. Control of the drug’s ionization degree using UV-Vis spectroscopy and NMR confirmed the presence of the deprotonated form at an alkaline pH (9.0–10.0). Under these conditions, the positively charged dendrimer molecule has 7.6% to 3.0% of protonated surface groups, respectively. Changes in the zeta potential of the complex observed in the presence of the drug in relation to the initial system are the effect of electrostatically immobilized drug molecules on the outer shell of the carrier structure. Based on changes in zeta potential, it was estimated that the 2 DOX molecules were located on the G4.0 PAMAM surface. In addition, based on changes in the absorbance intensity of the UV-Vis spectra, the binding of one to a maximum of ten DOX molecules in the structure is observed, depending on the pH of the environment and the molar ratio of the starting components. The strong effect of an alkaline pH on the efficiency of complex formation was determined. The data presented showed that the greatest number of DOX molecules bind when the components occur at pH 9.0 or 9.5 (*n* = 2) and the least at pH 10.0 (*n* = 1). Optimization of the pH of complex formation determined that pH 9.5 is most suitable for testing the effects of different DOX concentrations on binding efficiency. It was shown that the higher the DOX concentration, the higher the binding efficiency. Increasing the molar ratio to 1:24 resulted in an increase in the number of bound DOX molecules to n = 10. However, a concentration of doxorubicin (>0.25 mg/mL) that is too high is associated with strong systemic aggregation and DOX precipitation. Circular dichroism spectra analysis shows a stable dendrimer-drug complex after dialysis of all the systems obtained. The binding constant of the drug to the dendrimer under alkaline conditions shows higher drug affinity compared to literature data and is equal to *K*_9.0_ = (2.39 ± 0.43) × 10^2^ M^−1^, *K*_9.5_ = (2.73 ± 0.16) × 10^2^ M^−1^, and *K*_10.0_ = (3.98 ± 0.48) × 10^2^ M^−1^ for pH 9.0, 9.5, and 10.0, respectively. Circular dichroism spectra analysis shows a stable dendrimer-drug complex after dialyzation of all the systems obtained. In conclusion, the present study may contribute to the understanding of interactions in the G4.0 PAMAM-DOX system at the molecular level and thus help to find a representation of the mechanism of action and release of the active substance. Thus, they may bring closer the application of dendrimer-based nanotechnology in clinical practice.

## Figures and Tables

**Figure 1 pharmaceutics-15-00875-f001:**
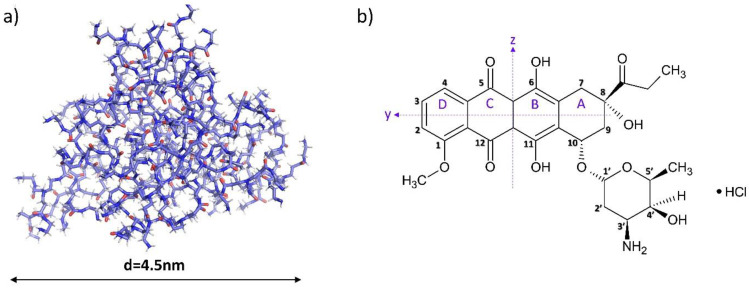
Structures of (**a**) G4.0 PAMAM dendrimer (The PyMOL Molecular Graphics System, Version 2.0 Schrödinger, LLC) and (**b**) doxorubicin hydrochloride (ACD/ChemSketch Freeware 2022.1.0).

**Figure 2 pharmaceutics-15-00875-f002:**
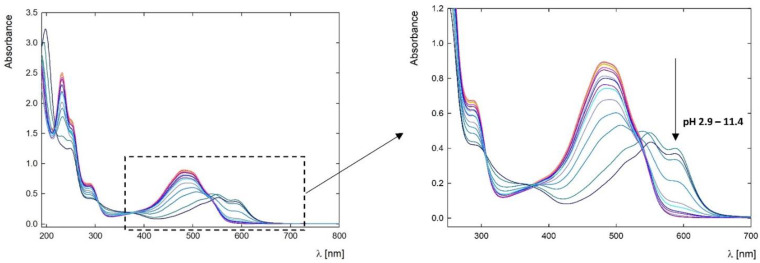
UV-Vis spectra of doxorubicin depend on the pH of the solution, with a concentration of c = 50 ppm in water in the pH range of 2.9–11.4.

**Figure 3 pharmaceutics-15-00875-f003:**
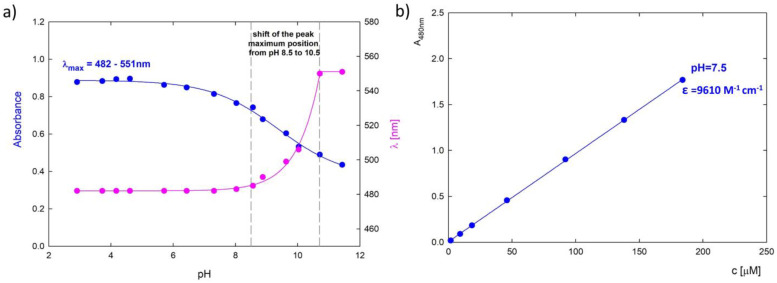
(**a**) A summary of the position of the absorbance maximum of the UV-Vis spectra for DOX (blue curve) and the shift of the absorbance maximum depending on the pH of the solution (pink curve); (**b**) Dependence of absorbance level on DOX concentration at pH 7.50 and wavelength *λ* = 480 nm.

**Figure 4 pharmaceutics-15-00875-f004:**
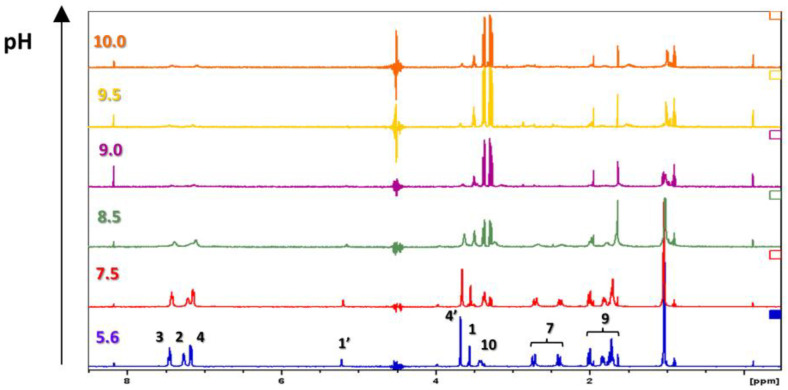
^1^H NMR spectra at different pH (c = 1.0 mg/mL, 50/50 vol% H_2_O/D_2_O).

**Figure 5 pharmaceutics-15-00875-f005:**
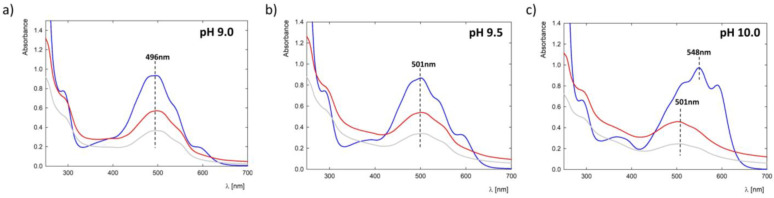
UV-Vis spectra of G4.0 PAMAM-DOX complexes with a molar ratio of 1:6 for selected pH values of the starting solution: (**a**) 9.0, (**b**) 9.5, and (**c**) 10.0 (blue line: starting complex, red: after 24 h of mixing, and gray: after dialysis).

**Figure 6 pharmaceutics-15-00875-f006:**
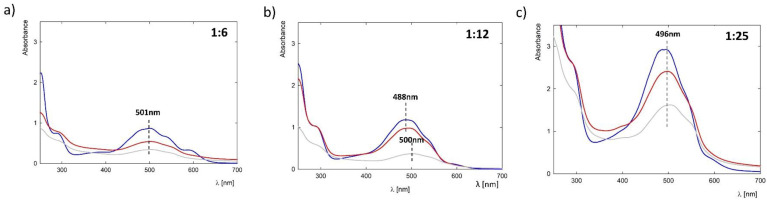
UV-Vis spectra of G4PAMAM-DOX complexes with pH 9.5 for selected G4PAMAM-DOX molar ratios: (**a**) 1:6, (**b**) 1:12, and (**c**) 1:24 (blue line: starting complex, red: 24 h of mixing, and gray: after dialysis).

**Figure 7 pharmaceutics-15-00875-f007:**
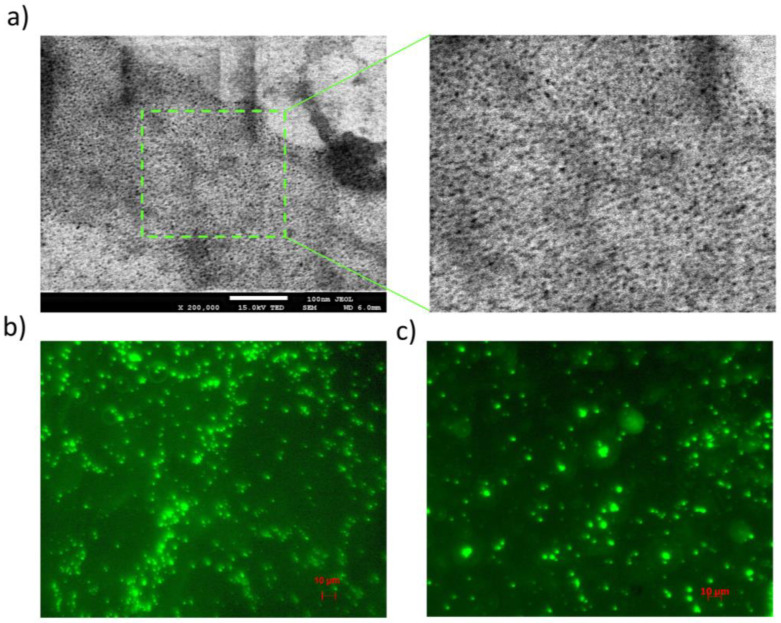
The visualization of the dendrimer structures of (**a**) the original G4PAMAM molecule using SEM; (**b**) the G4.0 PAMAM-DOX complex (1:6 at pH 9.5); and (**c**) the G4.0 PAMAM-DOX complex (1:12 at pH 9.5) from a fluorescence microscope.

**Figure 8 pharmaceutics-15-00875-f008:**
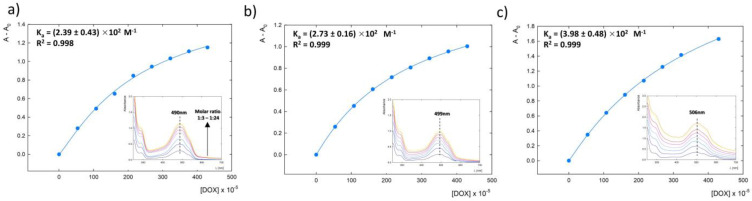
Relationship of *A* to *c_DOX_* for the G4.0 PAMAM-DOX complex for three pH values: (**a**) 9.0, (**b**) 9.5, and (**c**) 10.0.

**Figure 9 pharmaceutics-15-00875-f009:**
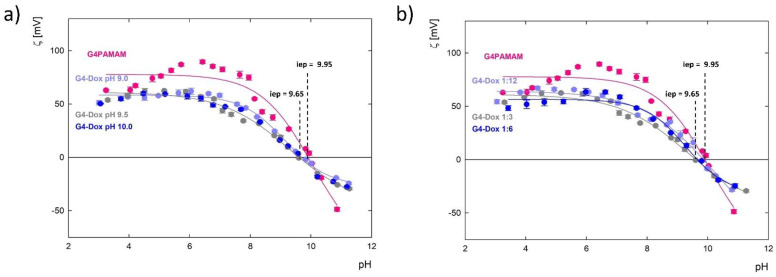
Change in the zeta potential (ζ) of the G4.0 PAMAM dendrimer and complexes with doxorubicin: (**a**) depending on the pH of formation and (**b**) the molar ratio of the components at pH 9.5.

**Figure 10 pharmaceutics-15-00875-f010:**
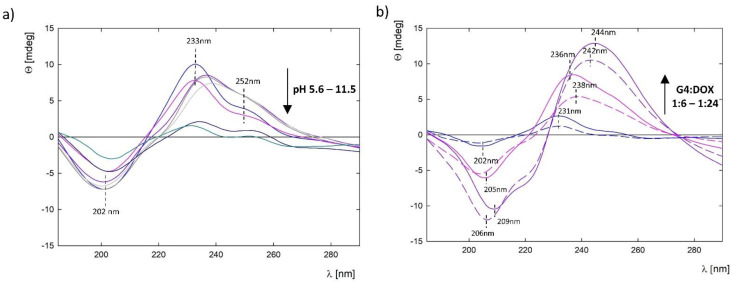
CD spectra for (**a**) DOX at a concentration of 0.1 mg/mL in the pH range of 5.6–11.5; (**b**) G4.0 PAMAM-DOX complexes depending on the molar ratio after 24 h of mixing (dark blue line: 1:6, pink line: 1:12, purple line: 1:24; solid lines: complex before dialysis, and dashed lines: complex after dialysis).

**Table 1 pharmaceutics-15-00875-t001:** Doxorubicin concentrations in G4.0 PAMAM-DOX 1:6 complexes after dialysis depending on the pH of complex formation in terms of the number of DOX particles per dendrimer molecule (N_DOX/PAMAM_), loading content (LC), and encapsulation efficiency (EE).

pH_i_	C_DOX_ [mg/mL]	N_DOX/PAMAM_	LC [%]	EE [%]
9.0	0.021	2.1	8.40	36.21
9.5	0.019	1.9	7.60	32.76
10.0	0.012	1.2	4.80	20.69

**Table 2 pharmaceutics-15-00875-t002:** Doxorubicin concentrations in G4.0 PAMAM-DOX complexes (pH = 9.5) after dialysis depending on the molar ratio of carrier/drug in terms of the number of DOX particles per dendrimer particle (N_DOX/PAMAM_), loading content (LC), and encapsulation efficiency (EE).

Molar Ratio	C_DOX_ [mg/mL]	N_DOX/PAMAM_	LC [%]	EE [%]
1:6	0.019	1.9	7.60	32.76
1:12	0.047	4.6	18.80	38.52
1:24	0.098	9.6	39.20	40.16

**Table 3 pharmaceutics-15-00875-t003:** Particle size (*d*) and polydispersity index (PDI) of G4.0 PAMAM-DOX complexes at different pHs and molar ratios of formation after dialysis, determined using the DLS method (analysis by intensity).

Initial Conditions	*d* [nm]	PDI
Influence of the pH on formation (molar ratio of 1:6)		
9.0	712.4 ± 77.2 (75%) 141.9 ± 14.4 (25%)	0.67
9.5	420.3 ± 55.3	0.36
10.0	531.2 ± 29.5	0.84
Influence of the molar ratio on formation (pH 9.5)		
1:6	420.3 ± 55.3	0.36
1:12	673.7 ± 35.8	0.79
1:24	136.1 ± 6.3	0.36

**Table 4 pharmaceutics-15-00875-t004:** Effective charge (*N_c_*), degree of ionization (*α*), and a number of ionized surface groups (*n*_NH3+_) for an aqueous solution of G4PAMAM dendrimer as a function of pH.

pH	*N_c_* [*e*]	*α_e_* [%]	*n* _NH3+_
4.0	8.4	13.1	8.9
5.0	10.2	16.0	10.3
6.0	11.5	18.2	11.7
7.0	11.4	17.8	11.4
8.0	7.9	12.4	7.9
9.0	4.9	7.6	4.0
10.0	-1.9	3.0	1.8

## Data Availability

Not applicable.

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
