# Peer review of "Effect of Alkaline Conditions on Forming an Effective G4.0 PAMAM Complex with Doxorubicin"

_pharmaceutics, 2023, doi:10.3390/pharmaceutics15030875_

Round 1
Reviewer 1 Report
The manuscript pharmaceutics-2209178 "Interaction of doxorubicin with G4.0 PAMAM dendrimer" by Szota and Jachimska describes the study of the effect of the PAMAM dendrimer ionization degree on doxorubicin (DOX) complexation. The obtained results have been confirmed by UV-Vis, NMR, fluorescence and CD spectroscopy. Despite the fact that the complexation of doxorubicin by PAMAM dendrimers has been studied earlier, this manuscript may be of interest to readers of Pharmaceutics.
Reviewer’s comments:
1. The title of the manuscript does not describe its content well. I recommend authors to choose a more specific title.
2. Many examples of the creation of PAMAM-DOX systems are available in the literature. The authors give only one example in the Introduction. First, it is necessary to pay attention to the results obtained earlier. Secondly, the authors need to write more clearly how their results differ from those previously obtained.
3. There are no titration spectra in the manuscript and supplementary materials. It is known that the calculation of titration is correct only when the stoichiometry of the complex is known. But there is no information about the complexation stoichiometry in the manuscript. The Benesi–Hildebrand method is well used for 1:1 complexes, so it is not suitable for the obtained PAMAM-DOX complexes. In modern supramolecular chemistry, BindFit software is used to estimate binding constants. I recommend the author to use Bindfit after establishing the stoichiometry of the complex.
4. Please provide DLS graphics by intensity (for G4-DOX at different pH). For the potential application of the resulting PAMAM-DOX systems, their high monodispersity is required. What is the meaning of the polydispersity index (PDI)?
5. Figure 8. 0.43×103 is not equal to 4.3×104. The colors in the signature are also mixed up. The equation to the straight lines should be added to the Figure 8.
6. I recommend the authors to strengthen the Introduction part about the synthesis of PAMAM dendrimers derivatives and their complexation properties. Recent articles on this topic should be added, e.g., 10.3390/ijms222111901, 10.3390/pharmaceutics14122748, 10.1080/09205063.2020.1827921.
7. Why were only complexes with a ratio of 1:6 studied by fluorescence spectroscopy?
8. SEM images are uninformative. Please add electronic microscopy images (high magnification) with clear boundaries of nanoparticles.
9. Lines 231-232. "Below pH 9.0, no drug was observed in the post-dialysis complex." It is known in the literature that PAMAM dendrimers derivatives are capable of DOX encapsulation at pH less than 9 (e.g. in PBS, 10.1016/j.xphs.2022.07.009). How can the authors explain their results, which contradict those previously known? What are the practical applications of the obtained PAMAM/DOX nanosystems that do not work under physiological pH conditions?
10. Line 112. Please correct the word "dendraimer". Please recheck text for errors.
Author Response
Point-by-point response to reviewer’s comments
Changes made to the manuscript are marked in blue
We would like to thank the Reviewers for their constructive comments, and the attention paid to our manuscript. Below are the detailed answers to the referees’ reports:
Reviewer 1: The manuscript pharmaceutics-2209178 "Interaction of doxorubicin with G4.0 PAMAM dendrimer" by Szota and Jachimska describes the study of the effect of the PAMAM dendrimer ionization degree on doxorubicin (DOX) complexation. The obtained results have been confirmed by UV-Vis, NMR, fluorescence, and CD spectroscopy. Despite the fact that the complexation of doxorubicin by PAMAM dendrimers has been studied earlier, this manuscript may be of interest to readers of Pharmaceutics.
1) The title of the manuscript does not describe its content well. I recommend authors to choose a more specific title.
Author reply:
Thank you for this suggestion. The title has been changed to “Effect of alkaline conditions on the formation of an effective G4.0 PAMAM complex with doxorubicin”.
2) Many examples of the creation of PAMAM-DOX systems are available in the literature. The authors give only one example in the Introduction. First, it is necessary to pay attention to the results obtained earlier. Secondly, the authors need to write more clearly how their results differ from those previously obtained.
Author reply:
As suggested, we have expanded the introduction to include a reference to articles on similar topics: “Dendrimers are more stable due to their unique cationic end groups, but this stability may increase hemolytic toxicity, especially with higher-generation PAMAM dendrimers (>G5 PAMAM) [12][14][16]. In addition, the nanosize of the dendrimers and the interaction with plasma proteins allow them to drive passive targeting through the effect of increased permeability and retention (EPR) [12]. The above-mentioned advantages resulting from the unique structure of dendrimers may play a key role in the controlled and targeted release of doxorubicin and thus reduce its cardiotoxicity. A DOX-conjugated G5-NH2 dendrimer-based carrier with a diamino butane core was shown by Kuruvilla et al. to increase anticancer activity and reduce DOX-induced cardiotoxicity in a mouse hepatocellular carcinoma (HCC) model [17]. Kaminskas et al. documented a similar effect in preparation based on a 5th-generation PEG-ylated polylysine dendrimer, which was characterized by good efficacy and showed a much lower risk of cardiotoxicity than in the case of the drug alone [18]. Researchers mainly focus on obtaining dendrimer-DOX complexes under physiological conditions, which are often conjugated to other ligands and/or linkers, such as amides, hydrazone, folic acid, hyaluronic acid or peptides [15][19][20][21]. Guo et al. Presented a combination therapy using doxorubicin and cisplatin conjugated with G4.0 PAMAM and hyaluronic acid, which showed high efficacy considering the inhibition of tumor growth as well as the reduction of DOX toxicity [21]. Zhong et al. demonstrated an equally high efficacy of DOX conjugate with carboxylated G4.0 PAMAM in local therapy of lung cancer [22][23]. Many studies show that dendrimers have great potential in active substance delivery and imaging. In contrast, despite this, they have yet to be tested in clinical trials for cancer therapy[15].
Despite the many studies on various dendrimer conjugates, the present study focuses on the interaction in the G4.0 PAMAM-DOX system at the molecular level. In the present study, special attention was paid to the effect of the degree of ionization of the components on the efficiency of complex formation, so a method of forming G4.0 PAMAM-DOX complexes under alkaline conditions was proposed. The following studies were focused on optimizing the conditions for the formation of the G4.0 PAMAM complex with doxorubicin, which can affect the subsequent application of nanosystems in vitro and the kinetics of drug release. To control the degree of ionization of the system and its stability, UV-Vis, NMR, and CD spectroscopy were used. The size and effective charge of the complex were monitored using DLS and electrophoretic mobility. Images from fluorescence microscopy confirmed the monomeric form of the drug in the structure of dendrimer complexes.
3) There are no titration spectra in the manuscript and supplementary materials. It is known that the calculation of titration is correct only when the stoichiometry of the complex is known. But there is no information about the complexation stoichiometry in the manuscript. The Benesi–Hildebrand method is well used for 1:1 complexes, so it is not suitable for the obtained PAMAM-DOX complexes. In modern supramolecular chemistry, BindFit software is used to estimate binding constants. I recommend the author to use Bindfit after establishing the stoichiometry of the complex.
Author reply:
Information on the stoichiometry of the complex is provided in Tables 1 and 2, where we report the following number of molecules bound to one dendrimer molecule:
Table 1. Doxorubicin concentrations in G4.0 PAMAM-DOX 1:6 complexes after dialysis depend on the pH of complex formation in terms of the number of DOX particles per dendrimer molecule (NDOX/PAMAM).
|
pHi |
CDOX [mg/mL] |
NDOX/PAMAM |
|
9.0 |
0.021 |
2.1 |
|
9.5 |
0.019 |
1.9 |
|
10.0 |
0.012 |
1.2 |
Table 2. Doxorubicin concentrations in G4.0 PAMAM-DOX complexes after dialysis depend on the molar ratio of complex formation in terms of the number of DOX particles per dendrimer molecule (NDOX/PAMAM).
|
Molar ratio |
CDOX [mg/mL] |
NDOX/PAMAM |
|
1:6 |
0.019 |
1.9 |
|
1:12 |
0.021 |
2.1 |
|
1:25 |
0.098 |
9.6 |
Stoichiometry was determined using equilibrium dialysis and UV-Vis spectroscopy. The results show that for most complexes the number of drug molecules bound to the dendrimer is about n=2, the exceptions are the G4.0PAMAM-DOX 1:6 complex at pH 10.0 where n=1.2 molecule was obtained, and G4.0PAMAM-DOX 1:24 at pH 9.5 where n=9.6 molecules were obtained.
As suggested, we estimated the binding constants using the BindFit software. However, this software allows the determination of binding constants for the following stoichiometries: 1:1, 1:2 or 2:1. We choose a 1:2 UV-Vis model was used for the complexes at pH 9.0 and 9.5, while a 1:1 UV-Vis model was used for the complex at pH 10.0. Using the Bindfit software, we do not observe pH influence on the binding constants values, which are 0.73×104 and 1.12×104 for the 1:1 and 1:2 models, respectively. Therefore, we decided to use Hill's method, is to determine the binding constants [1,2,3]:
= =
Where A (= Aobs - A0) is the change in absorbance, Amax is the maximum change of absorbance,
[L]0 is the initial guest concentration, n is the Hill coefficient, and Ka is the association constant.
Figure 8 has been replaced with titration spectra as a function of pH and curves of A - A0 vs. DOX concentration. The titration data were fit to this model using the nonlinear regression method within the SigmaPlot software. The table below shows the values of association constants and the number of DOX particles determined by Hill's method:
Table 3. Association constants (Ka) and Hill coefficients (n) are determined by the nonlinear fit of the titration data to the Hill equation for G4.0-DOX complexes at different pH of formation.
|
|
Ka [M-1] |
|
G4.0-DOX pH 9.0 |
2.39 ± 0.43 × 102 |
|
G4.0-DOX pH 9.5 |
2.73 ± 0.16 × 102 |
|
G4.0-DOX pH 10.0 |
3.98 ± 0.48 × 102 |
[1] T. P. Sheng, T. P. Sheng, X. X. Fan, G. Z. Zheng, F. R. Dai, and Z. N. Chen, “Cooperative binding and stepwise encapsulation of drug molecules by sulfonylcalixarene-based metal-organic supercontainers,” Molecules, vol. 25, no. 11, 2020, doi: 10.3390/molecules25112656.
[2] Hill, A.V. The possible effects of the aggregation of the molecules of hæmoglobin on its dissociation curves. J. Physiol. 1910, 40, iv–vii.
[3] Chenprakhon, P.; Sucharitakul, J.; Panijpan, B.; Chaiyen, P. Measuring Binding A_nity of Protein-Ligand Interaction Using Spectrophotometry: Binding of Neutral Red to Riboflavin-Binding Protein. J. Chem. Educ. 2010, 87, 829–831.
4) Please provide DLS graphics by intensity (for G4-DOX at different pH). For the potential application of the resulting PAMAM-DOX systems, their high monodispersity is required. What is the meaning of the polydispersity index (PDI)?
Author reply:
Below you can find a DLS particle size analysis by intensity for G4.0 PAMAM-DOX complexes as a function of pH of formation. G4.0 PAMAM dendrimers are characterized by high monodispersity. Their complexation with doxorubicin results in aggregation of the systems up to about from 136.1 ± 6.3 (G4.0-DOX 1:24, pH 9.5) to as much as 712 ± 77.2 nm (G4.0-DOX 1:6, pH 9.0) (Table 4). However, the aggregation process does not affect the effectiveness of the complexes in biological systems, as demonstrated by in vitro tests as demonstrated in the response to the 9th review (Figure 4,5).
Table 4. Particle size and polydispersity index (PDI) of G4.0 PAMAM-DOX complexes at different pH and molar ratio of formation, after dialysis determined using the DLS method (analysis by intensity).
|
Initial conditions |
d [nm] |
PDI |
|
Influence of pH of formation (molar ratio 1:6) |
||
|
9.0 |
712.4 ± 77.2 (75%) 141.9 ± 14.4 (25%) |
0.67 |
|
9.5 |
420.3 ± 55.3 |
0.36 |
|
10.0 |
531.2 ± 29.5 |
0.84 |
|
Influence of molar ratio of formation (pH 9.5) |
||
|
1:6 |
420.3 ± 55.3 |
0.36 |
|
1:12 |
673.7 ± 35.8 |
0.79 |
|
1:24 |
136.1 ± 6.3 |
0.36 |
Figure 1. Particle size distribution of G4.0 PAMAM-DOX complexes by intensity as a function of pH of formation: a) 9.0; b) 9.5; c) 10.0 and initial molar ratio of components: d) 1:6; e) 1:12; f) 1:24.
5) Figure 8. 0.43×103 is not equal to 4.3×104. The colors in the signature are also mixed up. The equation to the straight lines should be added to the Figure 8.
Author reply:
This section was removed from the manuscript due to the inadequacy of applying the Benesi-Hildebrand equation to the present systems, as suggested in the 3rd review.
6) I recommend the authors to strengthen the Introduction part about the synthesis of PAMAM dendrimers derivatives and their complexation properties. Recent articles on this topic should be added, e.g., 10.3390/ijms222111901, 10.3390/pharmaceutics14122748, 10.1080/09205063.2020.1827921.
Author reply:
We thank you for providing interesting literature items, and because the manuscript does not involve the topic of dendrimer derivatives, other equally interesting references have been cited to explain the complexation properties of PAMAM dendrimers. Accordingly, the following sentences have been added to the introduction section:
“A highly attractive feature of dendrimer molecules is the possibility of extensive modification of surface groups, and, thus, the ease of creating conjugates with selected ligands. In addition, dendrimers, due to their amphiphilic nature, can increase the solubility and bioavailability of both hydrophobic and hydrophilic drugs[12]. Considering the hydrophobic core of PAMAM dendrimers and the hydrophilic nature of the peripheral part, the structural analogy to micelles can be observed, which gives the ability to both ionic and hydrophobic binding of ligands[16][17]. Active molecules can be transferred in two ways using dendrimer systems, by immobilization on the surface structure or encapsulation inside. it. Dendrimers can transport ligands that are bound in a variety of ways, which can affect the stability of the system as well as release kinetics. These strategies include physical as well as chemical interactions. The first includes non-covalent association, hydrogen bonding, hydrophobic or electrostatic interactions, while the second involves covalent coupling of drugs to surface groups of dendrimers[18].”
7) Why were only complexes with a ratio of 1:6 studied by fluorescence spectroscopy?
Author reply:
We also visualized the complex at a molar ratio of 1:12, which was added to the manuscript (Figure 7c). Unfortunately, we do not have a picture of the complex at a molar ratio of 1:24. Due to the transfer of complexes for cytotoxicity assays, we also do not have the ability to replicate this result.
8) SEM images are uninformative. Please add electronic microscopy images (high magnification) with clear boundaries of nanoparticles.
Author reply:
The contrast and quality of the SEM image were increased to better visualize the G4.0 PAMAM dendrimer nanostructures:
Figure 3. The visualization of the dendrimer structures of (a) the original G4PAMAM molecule using SEM.
9) Lines 231-232. "Below pH 9.0, no drug was observed in the post-dialysis complex." It is known in the literature that PAMAM dendrimers derivatives are capable of DOX encapsulation at pH less than 9 (e.g. in PBS, 10.1016/j.xphs.2022.07.009). How can the authors explain their results, which contradict those previously known? What are the practical applications of the obtained PAMAM/DOX nanosystems that do not work under physiological pH conditions?
Author reply:
We thank the reviewer for the valid observation. The methodology that is used for the formation of complexes in the G4PAMAM-DOX system involves both the mixing of the components and the subsequent dialysis process. During dialysis, drug molecules not bound to the dendrimer are removed through a semipermeable membrane. Based on this, we determine the binding efficiency. The methodology that other researchers adopt for the same system does not take into account dialysis, and therefore the separation of unbound drug molecules. They also prepared complexes in buffers [1-4]. Thus, the authors assume a high binding efficiency of the drug to the PAMAM structure. As evidence of the failure to obtain an effective result at physiological pH in water, we include the following spectrum, which shows the absence of DOX molecules in the complex at pH 7.5 after dialysis:
Figure 2. UV-Vis spectra for G4.0 PAMAM-DOX complexes at molar ratio 1:3 and different pH of formation: 7.5 (green lines) and 10.0 (blue lines).
The pH of the complex formation does not affect its potential subsequent use. Alkaline conditions refer to the degree of ionization of the components and are needed only for the attachment of the drug to the dendrimer. In the case of complexes, after 24 hours of mixing and after the dialysis process, the pH of the systems spontaneously drops to a value of 7.0 - 7.5, so that the final formulations are at physiological pH.
In addition, to ensure that the G4.0 PAMAM-DOX complexes are working properly in vitro, we include the results of the viability tests. Below we present selected cytotoxicity results for the G4.0 PAMAM-DOX system and G4.0 PAMAM itself against lung cancer cells (NCL-H23) and human keratinocytes (HaCaT).
Figure 3. The cytotoxicity of G4PAMAM dendrimer (G4PAMAM-empty) and its complexes with DOX at different pH of formation: 9.0 (G4PAMAM-Dox-1), 9.5 (G4PAMAM-Dox-2) and 10.0 (G4PAMAM-Dox-3) against lung cancer cells and human keratinocytes cells after 24h and 72h of incubation.
Figure 4. The cytotoxicity of G4PAMAM dendrimer (G4PAMAM-empty) and its complexes with DOX prepared ad different molar ratios: 1:12 (G4PAMAM-Dox-1), 1:25 (G4PAMAM-Dox-2) and 1:50 (G4PAMAM-Dox-3) against lung cancer cells and human keratinocytes cells after 24h and 72h of incubation.
The presented results show a differentiated cellular response depending on both the method of complex preparation and the concentration of the drug in the formulation. With the visible effect of G4P.0 AMAM-DOX complexes, the lack of toxicity of the carrier itself is observed. At some concentrations, a slightly enhanced cytotoxic effect is also observed for the complexes compared to DOX alone (Figure 4b). In summary, the results represent the correct drug release from the dendrimer carrier in vitro. However, since the present results are still pending due to the obtaining of relevant statistics, they will be published in a subsequent manuscript.
[1] P. S. Lai et al., “Doxorubicin delivery by polyamidoamine dendrimer conjugation and photochemical internalization for cancer therapy,” J. Control. Release, 2007, doi: 10.1016/j.jconrel.2007.06.012.
[2] X. L. Guo et al., “Co-delivery of cisplatin and doxorubicin by covalently conjugating with polyamidoamine dendrimer for enhanced synergistic cancer therapy,” Acta Biomater., vol. 84, pp. 367–377, 2019, doi: 10.1016/j.actbio.2018.12.007.
[3] L. M. Kaminskas et al., “Doxorubicin-conjugated PEGylated dendrimers show similar tumoricidal activity but lower systemic toxicity when compared to PEGylated liposome and solution formulations in mouse and rat tumor models,” Mol. Pharm., vol. 9, no. 3, pp. 422–432, 2012, doi: 10.1021/mp200522d.
[4] P. Chanphai et al., “PAMAM dendrimers in drug delivery: Loading efficacy and polymer morphology,” Can. J. Chem., vol. 95, no. 9, pp. 891–896, 2017, doi: 10.1139/cjc-2017-0115.
10) Line 112. Please correct the word "dendraimer". Please recheck text for errors.
Author reply:
All errors have been corrected.

Reviewer 2 Report
· The abstract should be refined and shown the most important experiment results or it should be presented in a findings-oriented format in which the most important results and conclusions are summarized.
· In introduction section, I suggest to improve literature review to provide the background presentation and the application of this manner in cancer therapy. The authors should highlight the novelty of the manuscript and what is new in their work compared to other published literature.
· In results and discussion section, I recommend more profound discussion about the experiment results instead of the simple presentation about the experiment principle and results.
· In the method sections it would be useful to provide the accurate quantity of excipients used.
· Why do the authors think about using this present method? What is new? What is the aim of this work; what is the gap to cover?
· Cytotoxicity of the samples should be examined on a cancer and healthy cell line.
· conclusions can be expanded a little bit more to clearly state the best results obtained from this study and the optimal formulations.
Author Response
Point-by-point response to reviewer’s comments
Changes made to the manuscript are marked in blue
We would like to thank the Reviewers for their constructive comments, and the attention paid to our manuscript. Below are the detailed answers to the referees’ reports:
Reviewer 2
1) The abstract should be refined and shown the most important experiment results or it should be presented in a findings-oriented format in which the most important results and conclusions are summarized.
Author reply: As suggested, it was decided to remove the first two sentences “The beneficial therapeutic properties of doxorubicin (DOX) have been confirmed in many studies, but its wide application is limited by high cardiotoxicity. The solution to this problem may be using a nanocarrier, such as G4.0 PAMAM dendrimer in which DOX molecules can be bound by encapsulation or immobilization.” from the abstract and focus only on the results of the work. The abstract has been revised to the following form: “In this work, special attention was paid to the correlation between the degree of ionization of the components and the effective formation of the complex under alkaline conditions. Using UV-Vis, 1H NMR, and CD, structural changes of the drug depending on the pH were monitored. In the pH range of 9.0 to 10.0, the G4.0 PAMAM dendrimer can bind 1 to 10 DOX molecules, while the efficiency increases with the concentration of the drug relative to the carrier. The binding efficiency was described by the parameters of loading content (LC=4.80-39.20%) and encapsulation efficiency (EE=17.21-40.16%.), which values increased two- or even fourfold depending on the conditions. The highest efficiency was obtained for G4.0PAMAM-DOX at molar ratio 1:24. Nevertheless, regardless of the conditions DLS study indicates the system aggregation. Changes in the zeta potential confirm the immobilization of an average of two drug molecules on the dendrimer' surface. Circular dichroism spectra analysis shows a stable dendrimer-drug complex of all the systems obtained. Since the doxorubicin molecule can simultaneously act as a therapeutic and an imaging agent, the theranostic properties of the PAMAM-DOX system have been demonstrated by high fluorescence intensity observable on fluorescence microscopy.”
2) In introduction section, I suggest to improve literature review to provide the background presentation and the application of this manner in cancer therapy. The authors should highlight the novelty of the manuscript and what is new in their work compared to other published literature.
Author reply: As suggested, we have expanded the introduction to include a reference to articles on similar topics: “Dendrimers are more stable due to their unique cationic end groups, but this stability may increase hemolytic toxicity, especially with higher-generation PAMAM dendrimers (>G5 PAMAM) [12][14][16]. In addition, the nanosize of the dendrimers and the interaction with plasma proteins allow them to drive passive targeting through the effect of increased permeability and retention (EPR) [12]. The above-mentioned advantages resulting from the unique structure of dendrimers may play a key role in the controlled and targeted release of doxorubicin and thus reduce its cardiotoxicity. A DOX-conjugated G5-NH2 dendrimer-based carrier with a diaminobutane core was shown by Kuruvilla et al. to increase anticancer activity and reduce DOX-induced cardiotoxicity in a mouse hepatocellular carcinoma (HCC) model [17]. Kaminskas et al. documented a similar effect in preparation based on a 5th-generation PEG-ylated polylysine dendrimer, which was characterized by good efficacy and showed a much lower risk of cardiotoxicity than in the case of the drug alone [18]. Reseachers mainly focus on obtaining dendrimer-DOX complexes under physiological conditions, which are often conjugated to other ligands and/or linkers, such as amides, hydrazone, folic acid, hylaruonic acid or peptides [15][19][20][21]. Guo et al. Presented a combination therapy using doxorubicin and cisplatin conjugated with G4.0 PAMAM and hyaluronic acid, which showed high efficacy considering the inhibition of tumor growth as well as the reduction of DOX toxicity [21]. Zhong et al. demonstrated an equally high efficacy of DOX conjugate with carboxylated G4.0 PAMAM in local therapy of lung cancer [22][23]. Many studies show that dendrimers have great potential in active substance delivery and imaging. In contrast, despite this, they have yet to be tested in clinical trials for cancer therapy[15].
Despite the many studies on various dendrimer conjugates, the present study focuses on the interaction in the G4.0 PAMAM-DOX system at the molecular level. In the present study, special attention was paid to the effect of the degree of ionization of the components on the efficiency of complex formation, so a method of forming G4.0 PAMAM-DOX complexes under alkaline conditions was proposed. The following studies were focused on optimizing the conditions for the formation of the G4.0 PAMAM complex with doxorubicin, which can affect the subsequent application of nanosystems in vitro and the kinetics of drug release.To control the degree of ionization of the system and its stability, UV-Vis, NMR, and CD spectroscopy were used. The size and effective charge of the complex were monitored using DLS and electrophoretic mobility. Images from fluorescence microscopy confirmed the monomeric form of the drug in the structure of dendrimer complexes.
3) In results and discussion section, I recommend more profound discussion about the experiment results instead of the simple presentation about the experiment principle and results.
Author reply:
We have expanded some sections to include a broader interpretation, also we encourage to reread the manuscript as much of the results have changed due to the opinions of other reviewers.
4) In the method sections it would be useful to provide the accurate quantity of excipients used.
Author reply: In the methodology section, approximate volumes of NaOH added to the complexes to change the pH were added. It was not possible to give an exact volume because each drug concentration/volume of the system responds to pH changes differently, so these are not constant values.
5) Why do the authors think about using this present method? What is new? What is the aim of this work; what is the gap to cover?
Author reply: As many works show, dendrimers have great potential for use as carriers of active substances with therapeutic and theranostic applications, nevertheless there is still no commercially available preparation based on their technology. Therefore, optimal solutions that can solve this problem are being sought. What is new with regard to other available articles on similar topics is the precise determination of the degree of protonation of the components, especially doxorubicin, under the conditions of formation of complexes. The aim of our work was to optimize the formation conditions taking into account the anionic form of the drug, which occurs at a strongly alkaline pH. To date, most work has focused on the synthesis and action of these systems under physiological conditions, while few have considered the interaction in the carrier-drug system at the molecular level. This may be crucial in the subsequent mechanism of action and release kinetics of the active substance.
6) Cytotoxicity of the samples should be examined on a cancer and healthy cell line.
Author reply: We agree that cytotoxicity tests are extremely important in the study of present systems. We have performed in vitro tests for several cancer and normal cell lines. Nevertheless, these tests are in progress and will provide material for the next publication. Below we present selected cytotoxicity results for the G4PAMAM-DOX system and G4PAMAM itself against lung cancer cells (NCL-H23) and human keratinocytes (HaCaT).
Figure 3. The cytotoxicity of G4PAMAM dendrimer (G4PAMAM-empty) and its complexes with DOX at different pH of formation: 9.0 (G4PAMAM-Dox-1), 9.5 (G4PAMAM-Dox-2) and 10.0 (G4PAMAM-Dox-3) against lung cancer cells and human keratinocytes cells after 24h and 72h of incubation.
Figure 4. The cytotoxicity of G4PAMAM dendrimer (G4PAMAM-empty) and its complexes with DOX prepared ad different molar ratio: 1:12 (G4PAMAM-Dox-1), 1:25 (G4PAMAM-Dox-2) and 1:50 (G4PAMAM-Dox-3) against lung cancer cells and human keratinocytes cells after 24h and 72h of incubation.
Presented results show a differentiated cellular response depending on both the method of complex preparation and the concentration of the drug in the formulation. With the visible effect of G4P.0 AMAM-DOX complexes, the lack of toxicity of the carrier itself is observed. At some concentrations, a slightly enhanced cytotoxic effect is also observed for the complexes compared to DOX alone (Figure 4b). In summary, the results represent the correct drug release from the dendrimer carrier in vitro. Our next goal is to extend the present tests and show them in the next publication.
7) Conclusions can be expanded a little bit more to clearly state the best results obtained from this study and the optimal formulations.
Author reply: The conclusions have been revised and contain more specific information: “The study monitored the effect of the degree of ionization of doxorubicin on the effectiveness of complex formation with the G4.0 PAMAM dendrimer. Control of the degree of ionization of the drug using UV-Vis spectroscopy and NMR confirmed the presence of the deprotonated form at alkaline pH (9.0 - 10.0). Under these conditions, the positively charged dendrimer molecule has from 7.6% to 3.0% of protonated surface groups, respectively. Changes in the zeta potential of the complex observed in the presence of the drug in relation to the initial system are the effect of electrostatically immobilized drug molecules on the outer shell of the carrier structure. Based on changes in zeta potential, n=2 DOX molecules located on the G4.0 PAMAM surface were estimated. In addition, based on changes in the absorbance intensity of the UV-Vis spectra, the binding of one to a maximum of ten DOX molecules in the structure is observed, depending on the pH of the environment and the molar ratio of the starting components. The strong effect of alkaline pH on the efficiency of complex formation was determined. The data presented showed that the greatest number of DOX molecules bind when the components occur at pH 9.0 or 9.5 (n=2), and the least at pH 10.0 (n=1). Optimization of the pH of complex formation made it possible to choose pH 9.5 as suitable for testing the effect of DOX concentration on binding efficiency. It was shown that the higher the DOX concentration, the higher the binding efficiency. Increasing the molar ratio to 1:25 resulted in an increase in the number of bound DOX molecules to n=10. However, too high a concentration of doxorubicin (> 0.25mg/mL) is associated with strong aggregation of the system. Circular dichroism spectra analysis shows a stable dendrimer-drug complex after dialysis of all the systems obtained. The binding constant of the drug to the dendrimer structure strongly depends on pH and amounts to KpH 10.0 = 0.73 × 103 M-1, KpH 9.5 = 0.52 × 103 M-1 and KpH 9.0 = 0.43 × 103 M-1. The lowest affinity of doxorubicin for the dendrimer occurs at pH 10.0. Circular dichroism spectra analysis shows a stable dendrimer-drug complex after dialysis of all the systems obtained. In conclusion, the present study may contribute to the understanding of interactions in the G4.0 PAMAM-DOX system at the molecular level, and thus help to find a representation of the mechanism of action and release of the active substance. Thus, they may bring closer the application of dendrimer-based nanotechnology in clinical practice.”

Round 2
Reviewer 1 Report
I thank the authors for really improving the manuscript and answering my questions. However, some answers are incomplete. The reviewer’s comments and recommendations are below.
1) Please add DLS graphics by intensity to supplementary materials.
2) Usually, supramolecular systems with a high PDI (> 0.20) are considered polydisperse. The use of such systems for biomedical applications is undesirable. How can the authors explain the findings?
3) The authors write in line 162 that Bindfit software was used. But it writes below that the Hill method was used. Please correct this.
4) The cytotoxicity data against lung cancer cells (NCL-H23) and human keratinocytes (HaCaT) should be added to the manuscript. Without these results, it is difficult to understand the use of polydisperse systems obtained at different pH.
5) Unfortunately, the information content of SEM images has not improved. Please add a sample preparation procedure for SEM experiments. I also recommend changing something in this procedure, e.g., using metal spraying.
Author Response
We would like to thank the Reviewer for the constructive comments and the attention paid to our manuscript. Below are the detailed answers to the referees’ reports:
Reviewer: I thank the authors for really improving the manuscript and answering my questions. However, some answers are incomplete. The reviewer’s comments and recommendations are below.
1) Please add DLS graphics by intensity to supplementary materials.
Author reply: Particle size distributions have been added to the supplementary materials as Figure S1.
2) Usually, supramolecular systems with a high PDI (> 0.20) are considered polydisperse. The use of such systems for biomedical applications is undesirable. How can the authors explain the findings?
Author reply: It is assumed that nanoparticle systems intended for drug delivery in anticancer therapy should be monodisperse. The FDA does not provide criteria for the exact PDI range for nanoparticle-based controlled drug delivery systems [1]. Several systems have already been verified in the literature, which, despite their high polydispersity, are very biologically active [2-4]. Therefore, we have already conducted the first cytotoxicity tests, which positively verified the obtained G4.0 PAMAM complexes.
1.M. Danaei et al., “Impact of particle size and polydispersity index on the clinical applications of lipidic nanocarrier systems,” Pharmaceutics, vol. 10, no. 2, pp. 1–17, 2018,
- L. Wang et al., Reducing the cytotoxicity of poly(amindoamine) dendrimers by modification of a single layer of carboxybetaine, Langmuir, 2013, Vol 29, 28, pp 8914-8921,
- J. Markowicz et al., Biotin Transport-Targeting Polysaccharide-Modified PAMAM G3 Dendrimer as System Delivering -Mangostin into Cancer Cells and C. elegansWorms, Int. J. Mol. Sci. 2021, 22, 129
- J. Markowicz et al., Synthesis and Properties of Mangostin and Vadimezan Conjugates with Glucoheptoamidated and Biotinylated 3rd Generation Poly(amidoamine) Dendrimer, and Conjugation Effect on Their Anticancer and Anti-Nematode Activities, Pharmaceutics 2022, 14, 606
3) The authors write in line 162 that Bindfit software was used. But it writes below that the Hill method was used. Please correct this.
Author reply: Thank you for checking the manuscript in detail. The error has already been corrected: “UV-Vis spectroscopy performed the determination of pH-dependent binding constants of complex formation. A solution of DOX in molar ratios of 1:3, 1:6, 1:9, 1:12, 1:15, 1:18, 1:21, and 1:24 was added to a solution of G4.0 PAMAM with a constant concentration of 17.6 μM (0.25 mg/mL). UV-Vis spectra were measured in the wavelength range of 190–800 nm. Binding constant values were calculated using the Hill method.”
4) The cytotoxicity data against lung cancer cells (NCL-H23) and human keratinocytes (HaCaT) should be added to the manuscript. Without these results, it is difficult to understand the use of polydisperse systems obtained at different pH.
Author reply: Since the cytotoxicity texts are still in progress, we decided to add information on the preliminary results obtained, which show the applicability of polydisperse G4.0PAMAM-DOX systems in vitro:
“3.4 Size and stability analysis of G4-DOX complexes.
The size of the G4.0 PAMAM-DOX complexes depends on the condition of the complex formation and it was determined using the DLS method. The results are presented in Table 3. Particle size distribution of G4.0 PAMAM-DOX complexes by intensity are shown in Figure S1 in the supplementary materials. In all conditions, we observe the aggregation of dendrimer molecules in the presence of the drug in the system. The hydrodynamic diameter of the G4.0 PAMAM dendrimer at pH 10.0 equals d=4.9 ± 0.1nm at pH 10.0. The size of the aggregates ranges from 136.1 ± 6.3 (G4.0-DOX 1:24, pH 9.5) to as much as 712 ± 77.2 nm (G4.0-DOX 1:6, pH 9.0). Nevertheless, to verify the application use of the obtained polydisperse G4.0 PAMAM-DOX complexes, MTT tests were carried out, which showed the absence of cytotoxicity of the dendrimer itself and a decrease in cell viability after 72h incubation by 62% and 64% for lung cancer cells (NCL-H23) and human keratinocytes (HaCaT), respectively. The cytotoxicity of the drug itself under these conditions is comparable. The use of the carrier did not result in the drug’s effectiveness, but in vitro studies require further verification.
5) Unfortunately, the information content of SEM images has not improved. Please add a sample preparation procedure for SEM experiments. I also recommend changing something in this procedure, e.g., using metal spraying.
Author reply: There have been many attempts to use different ways to adsorb dendrimers onto grids (taking into account concentration, adsorption time, and presence of dye). Nevertheless, this method proved to be the most effective, and this is the best image we could get. A description of sample preparation for SEM imaging has been added to the methodology section as subsection 2.8:
“A drop of G4.0 PAMAM water solution with a concentration of 1×10-3 mg/mL was applied to the carbon grid (EM Resolutions Ltd., Keele, Staffordshire, England). After 1 minute, the excess solution was drained by gently touching the edge of the grid with a filter. The grid was dried in the air for 30 seconds. Then the dendrimers were stained by applying a drop of uranyl acetate solution with a concentration of 1% for 2 minutes. After this time, the excess dye was drained, and the prepared grid with dendrimers was allowed to dry. SEM studies were carried out using Field Emission Scanning Electron Microscope JEOL JSM-7500F equipped with an X-ray energy dispersive (EDS) system.”

Reviewer 2 Report
The corrections have been done completely and the manuscript can be accepted in present form.
Author Response

(The authors gave the same response as above.)

Round 3
Reviewer 1 Report
I thank the authors for answering all my questions.